# Neuroprotective–Neurorestorative Effects Induced by Progesterone on Global Cerebral Ischemia: A Narrative Review

**DOI:** 10.3390/pharmaceutics15122697

**Published:** 2023-11-29

**Authors:** Pedro Montes, Emma Ortíz-Islas, Citlali Ekaterina Rodríguez-Pérez, Elizabeth Ruiz-Sánchez, Daniela Silva-Adaya, Pavel Pichardo-Rojas, Victoria Campos-Peña

**Affiliations:** 1Laboratorio de Neuroinmunoendocrinología, Instituto Nacional de Neurología y Neurocirugía Manuel Velasco Suárez, Mexico City 14269, Mexico; 2Laboratorio de Neurofarmacología Molecular y Nanotecnología, Instituto Nacional de Neurología y Neurocirugía Manuel Velasco Suárez, Mexico City 14269, Mexico; emma.ortiz@innn.edu.mx (E.O.-I.); crodriguez@innn.edu.mx (C.E.R.-P.); 3Laboratorio de Neuroquímica, Instituto Nacional de Neurología y Neurocirugía Manuel Velasco Suárez, Mexico City 14269, Mexico; ruizruse@yahoo.com.mx; 4Laboratorio Experimental de Enfermedades Neurodegenerativas, Instituto Nacional de Neurología y Neurocirugía Manuel Velasco Suárez, Mexico City 14269, Mexico; adayadani@gmail.com; 5The Vivian L. Smith Department of Neurosurgery, The University of Texas Health Science Center at Houston McGovern Medical School, Houston, TX 77030, USA; pavel.s.pichardorojas@uth.tmc.edu

**Keywords:** global cerebral ischemia, dementia, neurorestoration, progesterone, neuroprotection

## Abstract

Progesterone (P4) is a neuroactive hormone having pleiotropic effects, supporting its pharmacological potential to treat global (cardiac-arrest-related) cerebral ischemia, a condition associated with an elevated risk of dementia. This review examines the current biochemical, morphological, and functional evidence showing the neuroprotective/neurorestorative effects of P4 against global cerebral ischemia (GCI). Experimental findings show that P4 may counteract pathophysiological mechanisms and/or regulate endogenous mechanisms of plasticity induced by GCI. According to this, P4 treatment consistently improves the performance of cognitive functions, such as learning and memory, impaired by GCI. This functional recovery is related to the significant morphological preservation of brain structures vulnerable to ischemia when the hormone is administered before and/or after a moderate ischemic episode; and with long-term adaptive plastic restoration processes of altered brain morphology when treatment is given after an episode of severe ischemia. The insights presented here may be a guide for future basic research, including the study of P4 administration schemes that focus on promoting its post-ischemia neurorestorative effect. Furthermore, considering that functional recovery is a desired endpoint of pharmacological strategies in the clinic, they could support the study of P4 treatment for decreasing dementia in patients who have suffered an episode of GCI.

## 1. Introduction

According to the World Health Organization, cardiovascular diseases are the leading cause of global mortality [1]. With advances in healthcare systems and the promotion of cardiopulmonary resuscitation (CPR) in emergency protocols, a high percentage of individuals may survive cardiac events. However, they often experience immediate or long-term neurological sequelae [2,3]. Along these lines, cardiac resuscitation has been associated with dementia [2,4]. This health problem has been further exacerbated by the comorbidity of cardiocerebrovascular alterations with coronavirus disease 2019 (COVID-19). This could lead to greater brain damage and subsequently increase the incidence of cognitive–emotional disorders characteristic of dementia [5,6,7].

The International Liaison Committee on Resuscitation (ILCOR), in their International Consensus on Cardiopulmonary Resuscitation and Emergency Cardiovascular Care Science with Treatment Recommendations (CoSTR), has emphasized the importance of addressing brain damage associated with cardiac arrest to improve neurological outcomes [8,9]. Therefore, the search for pharmacological or nonpharmacological treatments that target brain injury after cardiac arrest has been an area of interest in both basic and clinical research [10,11]. In experimental models of global cerebral ischemia (in the context of cardiac arrest), some pharmacologic agents have been shown to exert neuroprotective effects; however, these beneficial effects have not been observed in clinical trials [11]. Therapeutic hypothermia as a neuroprotective strategy after cardiac arrest is currently recommended by the American Heart Association, but it has the disadvantage that it must be implemented immediately [10]. Therefore, new neuroprotective/neurorestorative approaches are needed to treat brain damage associated with global ischemia. Endogenous compounds have been identified as possible candidates, including steroid hormones such as progesterone (P4) (Figure 1). P4 is a neurosteroid with diverse molecular/cellular actions in the central nervous system (CNS) throughout development and into aging [12,13,14]. Its potential neuroprotective/neurorestorative against brain ischemia is based on its pleiotropic capacity. This feature allows it to modulate various pathophysiological mechanisms and promote plastic recovery processes after ischemic events [12,13,14,15,16]. The present review aims to summarize and analyze reports documenting both neuroprotective and neurorestorative effects against morphofunctional damage resulting from global cerebral ischemia. To begin, a brief description of the pathophysiology of global ischemia is developed, along with an overview of the recommended guidelines for researching pharmacological agents and the specific attributes of P4 that make it a promising candidate for the treatment of this neurological condition.

## 2. Pathophysiology of Global Cerebral Ischemia and Morphofunctional Changes

During an episode of global cerebral ischemia, the blood flow is temporarily reduced to less than 25% of normal levels (which ranging from 50 to 100 mL/100 g brain tissue per minute). This reduces the supply of oxygen and glucose supply to the brain [17,18]. Reperfusion after global ischemia induces hemodynamic changes, such as hyperemia–hypoperfusion, and ultimately restoration of normal blood flow [19]. In addition, blood flow in the cerebral microcirculation is heterogeneous, exhibiting areas of no flow, low flow, and increased flow [19,20]. Beyond systemic and hemodynamic changes, an ischemic episode activates several pathophysiological mechanisms that may lead to cell death (Figure 2). The decrease in oxygen and glucose supplies may lead to ATP depletion, which disrupts energy-dependent ion pumps, resulting in altered cellular ion homeostasis, anoxic depolarization, and the massive release of excitatory neurotransmitters such as aspartate and glutamate from the presynaptic terminals [17,21,22,23]. The excitotoxicity of glutamate-mediated through N-methyl-D-aspartate receptors (NMDARs) induces an elevated Ca^2+^ permeability, thus amplifying intracellular Ca^2+^. Elevated cytosolic Ca^2+^ levels can activate numerous proteases such as caspases, calpains, lipases, and endonucleases [22,23,24,25,26]. Furthermore, ischemia–reperfusion increases oxidative stress due to the generation of reactive oxygen and nitrogen species (ROS/RNS), and activates neuroinflammatory mechanisms. Molecular damage mechanisms cause, in turn, neuronal alterations such as cytotoxic cellular edema, mitochondrial dysfunction, damage to the microtubules that make up the cytoskeleton, and various cytostructural alterations that can ultimately lead to cell death [23,25,27,28,29].

Neuronal death or injury commonly occurs in brain structures susceptible to global ischemia, and its magnitude is dependent on the severity and duration of the ischemic period. Vulnerable regions include pyramidal neurons of cornu ammonis (CA) of the hippocampus, especially those in sector CA1, pyramidal neurons of layers III and V of the cerebral cortex, small and medium-sized spiny neurons of the caudate nucleus and those of the putamen, as well as the Purkinje cells of the cerebellum [2,11,17,30].

Simultaneous to the pathophysiological mechanisms, an episode of ischemia also activates processes that can promote neuronal survival. Among them are the antioxidant, antiapoptotic, and anti-inflammatory mechanisms [31,32]. Furthermore, dynamic neural changes occur as a result of the brain’s plastic capacity to cope with ischemia. Structural changes such as a reduction in arborization and dendritic spines, as well as ultrastructural alterations of their synapses, have been demonstrated in the surviving neurons [33,34,35]. Neuronal death and/or such morphological changes in the remaining neurons of the hippocampus and the cerebral cortex are consistently related to cognitive–emotional impairments both in clinical observations and experimental models [2,17,36,37,38,39], suggesting a post-ischemia maladaptive plastic response [40,41]. However, other neural changes have also been found, including dendritic sprouting, spinogenesis, and neurogenesis. Since these post-ischemic plasticity processes may favor the reorganization of altered synaptic circuits and are compatible with morphofunctional recovery, they are called adaptive plastic changes [40,41,42,43]. Understanding the different processes, both brain injury and plasticity, has been the conceptual basis for designing the strategies to be followed in the studies that aim to evaluate possible treatments against an episode of cerebral ischemia.

## 3. Guide Recommendations for the Study of Potential Treatments against Global Cerebral Ischemia

Based on the temporal course of the pathophysiological mechanisms triggered by global cerebral ischemia, it was proposed that there is a “window of therapeutic opportunity” of a few hours in which such mechanisms can be countered through the administration of drugs or nonpharmacological strategies, and thus reduce neuronal death [44,45]. Under this neuroprotective strategy, in basic studies, some pharmacological agents have prevented or decreased ischemic brain injury; for example, Ca^+2^ channel blockers, anticonvulsants, drugs that reduce the release of excitatory neurotransmitters, glutamate receptor antagonists, GABA receptor agonists, antioxidant agents, and antiapoptotic drugs. No significant neuroprotection was found when translated to the clinic [45,46,47,48]. To overcome this discrepancy and achieve a better therapeutic efficacy of potential neuroprotective drugs, it has been suggested to consider some strategic recommendations [44,47]. Among the suggestions, it has been proposed that the neuroprotective drug to be studied has the potential to counteract several of the pathophysiological mechanisms activated by ischemia. Alternatively, a combination of various drugs can be employed against distinct pathophysiological mechanisms [44,46]. It is also recommended that the evaluation of the treatment be carried out in the long term to elucidate the final effect of the drug, since in short-term studies it is not known whether the neuroprotector is only delaying ischemic damage [44,49]. In addition, the neuroprotective effect of a drug should manifest itself as a reduction in long-term morphofunctional alterations caused by ischemia, which could not be compensated by the processes of spontaneous post-ischemia plasticity [50,51]. Once this endogenous neurorestorative capacity of the brain has been considered, it has also been proposed that the drugs to be evaluated have the ability to stimulate one or more of the post-ischemia plasticity mechanisms and, ideally, that the agent has the potential both to oppose the mechanisms of injury and to promote plastic recovery mechanisms [44,46,52]. Therefore, current recommendations promote the search for therapeutic strategies that have neuroprotective/neurorestorative effects [38,41]. Likewise, the study of treatments specifically oriented to the promotion of neurorestoration has been proposed, since its therapeutic time window is highly heterogeneous, ranging from days to weeks or even longer periods [53,54]. In concordance with current guides, neuroactive steroids are potential candidates to be considered, given their pleiotropic effects, for the treatment of brain pathologies including damage caused by an episode of global ischemia [55,56].

## 4. Progesterone as a Neuroactive Steroid and Its Mechanisms of Action

Steroid hormones, such as androgens, estrogens, and progestogens, come from peripheral glands. While they regulate reproductive functions, they also influence various processes in the CNS during both developmental and adult stages. For this reason, they are referred to as neuroactive steroids [12]. During development, steroid hormones exert their actions on neurons and glial cells by regulating survival, differentiation, and connectivity of specific neural groups. In adulthood, these neuroactive steroids regulate neuronal morphology and functionality by promoting the growth of their projections, synaptogenesis, neurotransmitter synthesis, and receptor expression for neurotransmitters, thus regulating synaptic transmission [12,57]. Neuroactive steroids also include locally synthesized steroid hormones in the brain, as well as exogenously administered synthetic steroids reaching the brain which may be used in clinical practice [12,58]. P4 synthesized in the brain or transported via the bloodstream is biotransformed to its reduced metabolites 5α- and 5β-dihydroprogesterone (DHP or pregnandione) by the action of 5α- or 5β-reductase, respectively, and to the four corresponding tetrahydroderivatives: 3α,5α-, 3β,5α-, 3α,5β-, and 3β,5β-tetrahydroprogesterone (pregnanolones) by 3α- or 3β hydroxysteroid oxidoreductase, respectively. The main biotransformation product of P4 and with greater effects on CNS is 3α,5α-tetrahydroprogesterone (allopregnanolone) [14,59].

The classic (or genomic) mechanism of action of steroids in the CNS, once they cross the cell membrane due to their lipophilic structure, is through interaction with their intracellular receptors. In particular, the P4 (PR) receptors that have been characterized mainly are the PR-A and PR-B isoforms. Active PRs are transcription factors that bind to P4 response elements, which are specific DNA sequences within the regulatory regions of target genes, thus regulating their genetic expression. PRs may increase or decrease the expression of a gene depending on joint action with other transcription factors and coactivators/corepressors, as well as epigenetic mechanisms according to the specific cellular microenvironment [14,15,16]. In the so-called nongenomic or rapid mechanism of action of P4, the hormone interacts with a variety of membrane receptors such as progesterone receptor membrane component 1 (PGRMC1) and membrane receptors (mPRs), of which the following five isoforms have been characterized: mPRα, mPRβ, mPRγ, mPRδ, and mPRε. Activation of mPRs by P4 mainly results in transduction of intracellular signaling pathways through second messengers, although it may also indirectly induce regulation of gene expression [14,16]. Another mechanism of P4 is the allosteric modulation of ionotropic receptors in the CNS. The positive modulation of GABA receptors by their metabolite allopregnanolone is one of the most important for its physiological relevance by decreasing neuronal excitability very quickly [13,14].

The described mechanisms of action of P4 underlie their molecular/biochemical effects, which in turn support their neuroprotective/neurorestorative potential for brain injury, including that caused by ischemia [13,15].

## 5. Biochemical and Cellular Effects Supporting Progesterone as a Treatment against Global Cerebral Ischemia

P4 may oppose ischemic pathophysiological mechanisms (Figure 2) since it inhibits neuronal excitability and attenuates intracellular calcium concentration by increasing GABAergic neurotransmission and decreasing excitotoxic glutamatergic neurotransmission [58,60,61,62]. Moreover, P4 increases the expression of antiapoptotic genes and/or decreases the expression of proapoptotic genes [63,64], exerts antioxidant effects [65], regulates glial reaction and decreases the expression of proinflammatory cytokines [66,67], decreases cerebral edema [68], and protects the integrity of the BBB [69,70].

Simultaneously, P4 induces effects that may favor the development of post-ischemic brain plasticity processes associated with adaptive cerebral reorganization (Figure 3). It has been reported that it promotes axonal myelination [70], decreases the neuritic growth inhibitor Nogo-A [71], regulates cytoskeletal proteins involved in brain plasticity [72], and increases the expression of growth-associated protein 43 (GAP-43) and synaptophysin in CA1, demonstrating that the hormone induces synaptogenesis in this region of the hippocampus [73]. P4 also modulates the expression of neurotrophic factors such as brain-derived neurotrophic factor (BDNF), vascular endothelial growth factor (VEGF), and neudesin (neuron-derived neurotrophic factor, NENF) [74,75], induces compensatory changes in neuronal cytoarchitecture [76], and regulates hippocampal neurogenesis [42,74,77].

Integrally, by reducing damage mechanisms or regulating post-ischemic plasticity processes, P4 has the potential to induce morphological changes in association with functional recovery. The latter is of great importance since it is a requirement for a neuroprotective/neurorestorative agent to be considered of potential utility in clinical practice [52].

## 6. Morphological and Functional Effects Induced by Progesterone on Global Cerebral Ischemia

The study of P4 as a neuroprotective agent has been of historical interest. For this reason, its pharmacological properties have been examined in various models of brain diseases. For ischemic lesions, studies have shown that P4 exerts neuroprotective effects in animals submitted to focal ischemia, reducing cerebral infarct volume and neurological deficits of these subjects [68,78,79,80,81,82,83,84,85,86]. Moreover, treatment with allopregnanolone, after focal ischemia, attenuates BBB dysfunction [69] and reduces cortical infarct volume [87]. The feedback between focal (stroke-related) cerebral ischemia and global (cardiac-arrest-related) cerebral ischemia has been crucial due to their similarities in some pathophysiological mechanisms. However, given that the morphofunctional consequences are different, current perspectives indicate that preclinical studies should be analyzed, considering this fact to differentially support the therapeutic application of a drug in the clinical setting [88,89].

Studies examining the therapeutic potential of P4 have been carried out in variant models of global ischemia, in different species, in varied administration temporal schemes, under different doses and routes of administration, and with evaluations at different times.

Treatment with P4 (10 mg/kg/day s.c.), seven days before and seven days after a 15 min episode of global ischemia induced by cardiorespiratory arrest reanimation in ovariectomized cats, decreases neurological disorders and significantly preserves both the population of pyramidal neurons in regions CA1, CA2, CA3, and hippocampal hilus as the population of small neurons of the caudate nucleus, 14 days post-ischemia [90,91]. In male mice exposed to 10 min of global ischemia by bilateral occlusion of the common arteries, P4 treatment (15 mg/kg i.p.) at 30 min pre- and at 24, 48, and 72 h post-ischemia decreases the immediate alterations of the memory, motor coordination, anxiety, seizure susceptibility, cerebral infarct size, oxidative stress, and the proinflammatory cytokine TNF-α [92]. P4 (15 mg/kg i.p.) administered once a day for 15 days before and 14 days subcutaneously after a 15 min episode of ischemia by the four-vessel occlusion (4-VO) model in male rats is also effective since it favors the motor function in relation to a significant preservation of CA1 and an increase in synaptogenesis in this hippocampal region [73].

P4 treatment (8 mg/kg i.v.) specifically in the post-ischemia period at 20 min, 2, 6, and 24 h after a 15 min episode of global cerebral ischemia induced by the 4-VO model in male rats significantly preserves the neuronal population of the CA1 and CA2 subfields of the hippocampus and prevents cortical shrinkage 21 days after ischemia [93]. In rats subjected to the same ischemic model for 13 min and subsequently treated with P4 (8 mg/kg i.v.) at 15 min, 2 h, 6 h, 24 h, 48 h, and 70 h, are also found partial neuronal preservation and reduction in DNA fragmentation in CA1 at 7 days post-ischemia [63]. The cognitive performance of P4-treated ischemic rats (with a similar dose and temporal schedule of administration) was evaluated in another study, after 14 days of a 15 min global ischemia episode. P4 was found to promote successful spatial learning and memory performance, which is associated with a decrease in the expression of molecules involved in the signaling pathway of Nogo A, a neurite growth inhibitor, in CA1; thus, the hormone would have favored a microenvironment compatible with a compensatory reorganization of remaining neurons in this hippocampal region [71].

Long-term morphofunctional effects of P4 have been analyzed after a more severe global ischemia episode by the 4-VO model. P4- or allopregnanolone-treated rats (8 mg/kg i.v.), at 20 min, 2, 6, 24, 48, and 72 h after 20 min of ischemia, have better functional performance in learning and memory tests compared to vehicle-treated rats 90 days after ischemia despite the fact that the loss of pyramidal neurons in the CA1, CA2, CA3, and hilus regions of the hippocampus is similar in both experimental groups, in addition to having no changes in the medial prefrontal cortex (mPFC) with respect to an intact group [56]. The analysis of the cytoarchitectonic characteristics of the remaining CA1 pyramidal neurons in the ischemic P4-treated rats revealed that they have sinuously branched dendrites with a similar number of bifurcations and whole density of dendritic spines, as well as higher proportional density of mushroom spines than those of the group of intact rats. P4 thus favors the brain’s intrinsic capacity to recover and adapt after a severe ischemic episode [76]. In a similar experimental protocol, it was found that P4 is capable of inducing adequate behavioral response to solve learning spatial and memory tasks 21 days after an even more severe episode of global ischemia (30 min). This functional recovery was also mainly related to the effects of P4 on post-ischemia plasticity to compensate for the severe loss of hippocampal pyramidal neurons. P4 treatment increased the survival of new mature neurons in the hippocampal dentate gyrus [42]. It is known that once they are integrated into pre-existing neural circuits, these new neurons may participate in the processing of spatial learning and memory [94,95].

On the other hand, P4 neuroprotective effects were also shown when male rats were exposed to chronic stress, an environmental factor that exacerbates neuronal damage and inflammation, prior to the global ischemia episode by the 4-VO model. Hormone treatment (8 mg/kg i.p.) at 2 h after ischemia followed by subcutaneous administrations at 6 h and once every 24 h for 7 days was able to significantly preserve the CA1 neuronal population. This decrease in brain damage was related to the regulation of the ischemic inflammatory environment by P4, since it decreased the activation of the NOD-like receptor pyrin domain containing 3 (NLRP3) inflammasome in the microglia, thus decreasing the activation of these cells while increasing their autophagy. Moreover, P4 decreased M1 microglia and expression of associated proinflammatory cytokines, and increased the level of expression of anti-inflammatory cytokines and neurotrophic factors in the hippocampus [67,96]. Given the relevance of the comorbidity of stress disorders with global cerebral ischemia, it would be interesting to carry out further studies that include long-term cognitive evaluations.

The protective potential of progesterone in combination with other hormones has also been investigated. Coadministration of P4 (8 mg/kg i.p.) and the antioxidant hormone melatonin (10 mg/kg i.p.), immediately after reperfusion, at 5 h and at 1 day–7 days in rats subjected to global ischemia during 20 min by bilateral occlusion of the common arteries, improves spatial learning and memory at 22 days after the ischemic episode in relation to an increase in BDNF and GDNF and high neuronal preservation in CA1 [97]. Since the morphofunctional preservation was significantly better with the coadministration of hormones compared to that induced by each of them, it would be important to analyze this combination in other models of more severe global ischemia. In another short-term study without morphofunctional analysis, it was found that combined treatment of P4 (10 mg/kg i.p.) and estradiol (0.1 mg/kg i.p.), daily for one week prior to ischemia and at 24, 48, and 72 h after 10 min ischemic episode by 4-VO, decreased oxidative stress in the hippocampus, striatum, and cortex of ovariectomized rats. However, an additive effect was not observed [65].

## 7. Discussion and Conclusions

The examination of progesterone’s neuroprotective and neurorestorative effects on global cerebral ischemia has thus far been conducted in animal models. The main findings summarized here show that P4 treatment, prophylactic and/or after an episode of global cerebral ischemia, effectively preserves the neuronal population of vulnerable brain structures in relation to functional recovery. This morphofunctional preservation shows that P4 has neuroprotective properties but is limited to the aforementioned conditions, since studies administering progesterone following a severe ischemic episode, involving a greater challenge, find morphological alterations due to high neuronal death. However, post-ischemia P4 administration maintains its capacity to induce functional recovery through long-term neurorestorative processes that compensate for neuronal loss (Figure 4). It is important to note that the elucidation of these neurorestorative effects of progesterone was reported in studies that followed the most recent strategic recommendations, such as analyzing post-severe ischemia treatment and carrying out, long-term, both morphological analysis and cognitive performance [44,47,53,54]. This pharmacological intervention more accurately replicates the clinical scenario, as patients with certain cardiovascular conditions that entail a period of severe global cerebral ischemia receive brain medical care when achieving comprehensive morphofunctional preservation is no longer feasible [2,3,4]. Therefore, future basic research should consider the targeted-plasticity strategy to promote post-ischemia neurorestoration by progesterone treatment (Figure 4).

The central nervous system experiences continuous changes, even in response to alterations resulting from ischemic events, via plastic processes throughout life [38,98]. Due to its pleiotropic nature, the hormone progesterone regulates multiple mechanisms of brain plasticity [55,56,72]. Consequently, the therapeutic intervention with progesterone to promote neurorestoration could be investigated within a considerably broader time frame, spanning at least several months post-ischemia [53,54], in contrast to the more limited timeframe typically considered for the neuroprotective approach [44,45]. The exploration and analysis of treatments targeting neurorehabilitation are crucial [99,100], as numerous treatments focused solely on neuroprotection have proven ineffective in clinical trials [45,46,47,48].

Studies examining the protective potential of progesterone in the context of global cerebral ischemia with comorbidities, as well as its potential when combined with other hormones, are still limited. Future research directions under these paradigms should not only focus on analyzing the neuroprotective approach but also take into account the neurorestorative approach.

Based on the results obtained in the basic studies, the administration of P4 has already been translated for its evaluation in clinical studies aimed at the treatment of cerebral alterations such as traumatic brain injury [101,102,103]. While these studies have reached the clinical trial III, and it has been suggested that additional investigation be carried out to conclude the discussion of the controversies [103,104], for the treatment of ischemic events its analysis has not yet begun. Basic studies presented here show that P4 treatment, regardless of the final magnitude of ischemic brain damage, complies with a major endpoint of neuroprotective/neurorestorative strategies by favoring functional recovery, which is a primary requirement expected in the clinical context. Consequently, its examination could be contemplated in a clinical trial, given the need worldwide to find potential therapeutic alternatives to curb the dementia epidemic linked to cardiocerebrovascular alterations [2,4].

## Figures and Tables

**Figure 1 pharmaceutics-15-02697-f001:**
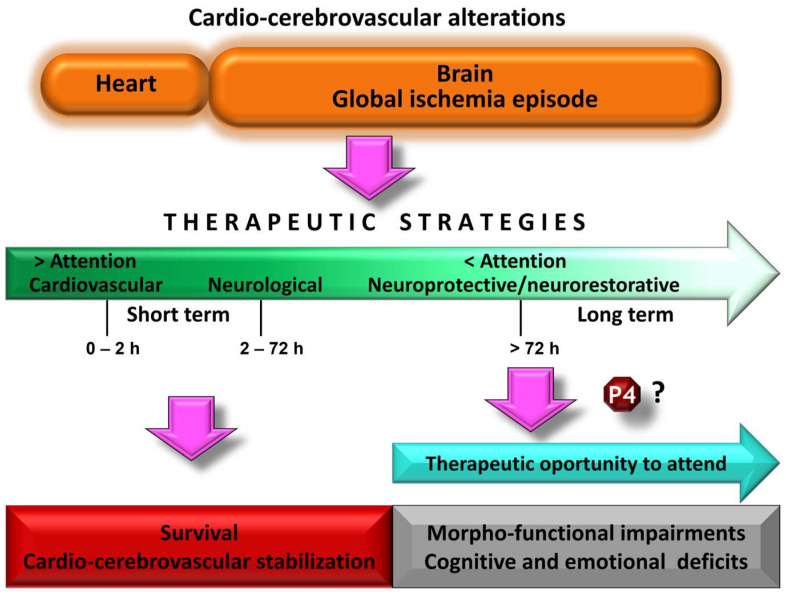
Schematic representation of the therapeutic attention after a cardiocerebrovascular disorder. The clinical urgency causes the cardiovascular component to be treated and stabilized as a priority. Progress in this area has contributed to improved survival rates. However, less attention to brain damage resulting from the global ischemic episode is related to an increase in alterations associated with dementia. It is time to move forward in its therapeutic care to lessen this trend. Neuroactive steroids such as progesterone (P4) could be therapeutic agents to be considered, based on the evidence shown in experimental models of global cerebral ischemia.

**Figure 2 pharmaceutics-15-02697-f002:**
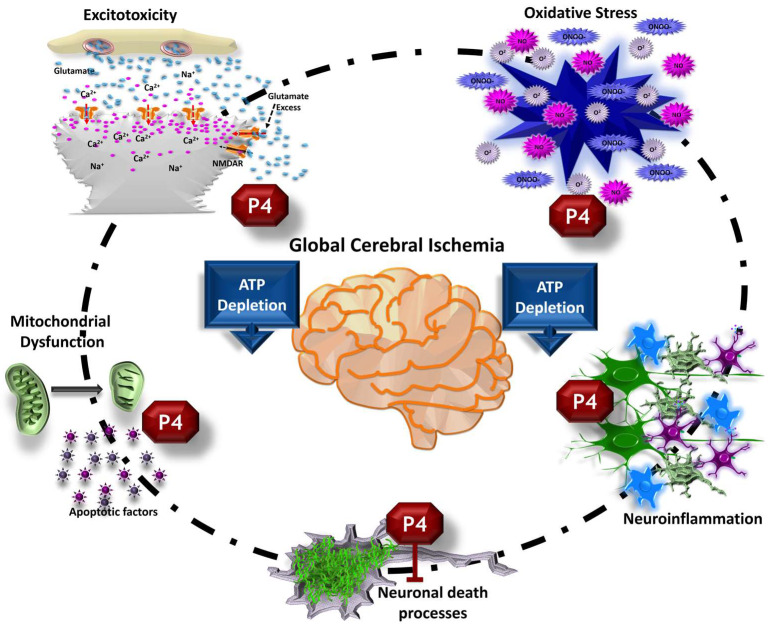
Effects of progesterone (P4) on the pathophysiology of global cerebral ischemia. P4 administration modulates various damage mechanisms, including glutamatergic excitotoxicity, mitochondrial dysfunction, oxidative stress, and neuroinflammatory pathways. Consequently, it has the potential to prevent or decrease post-ischemic neuronal death.

**Figure 3 pharmaceutics-15-02697-f003:**
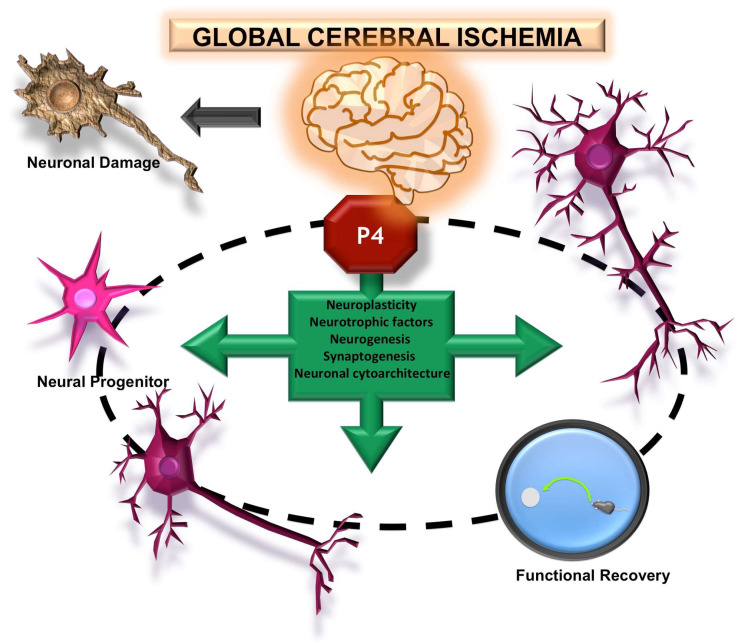
Effects of progesterone (P4) on brain plasticity after global ischemia. P4 treatment promotes the recovery of surviving neurons post-ischemia, inducing the restoration of their dendritic arborization and synaptic capacity. Furthermore, it supports the maturation of new neurons. These morphological changes are associated with improved cognitive–behavioral performance.

**Figure 4 pharmaceutics-15-02697-f004:**
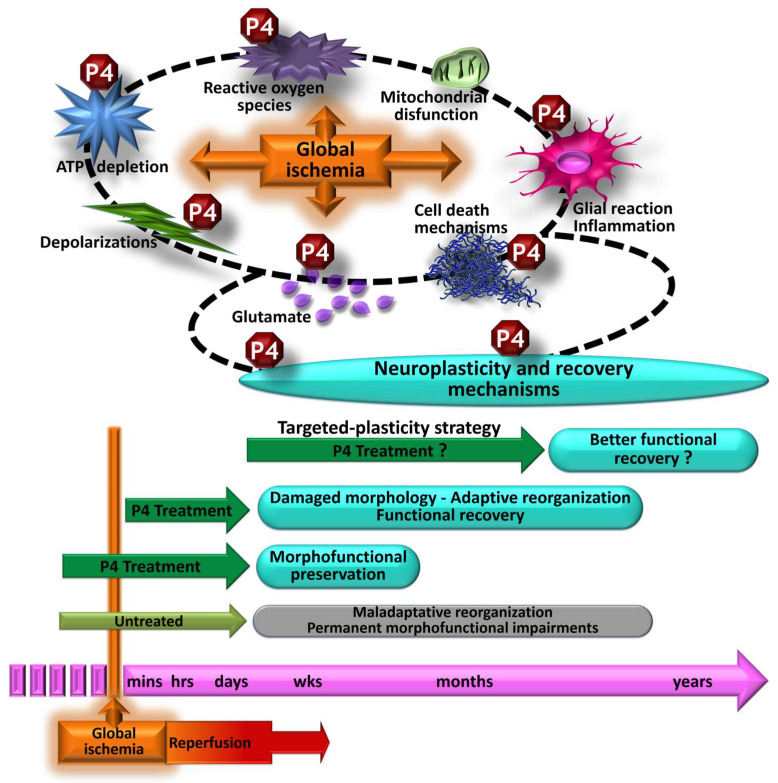
Morphofunctional effects of progesterone (P4) treatment against an episode of global cerebral ischemia. Treatment with progesterone pre- and post-ischemia regulates several pathophysiological mechanisms; thus, it is related to significant morphofunctional preservation. P4 treatment after severe ischemia (greater challenge) does not prevent morphological alterations. However, it promotes the development of adaptive plasticity processes, which in the long term are also related to functional recovery. Given the clinical relevance, it would be important to investigate whether, with specific treatment schemes to favor restorative post-ischemia plasticity mechanisms, progesterone promotes better functional recovery.

## Data Availability

Data sharing is not applicable.

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
