# Peer review of "Neuroprotective–Neurorestorative Effects Induced by Progesterone on Global Cerebral Ischemia: A Narrative Review"

_pharmaceutics, 2023, doi:10.3390/pharmaceutics15122697_

Round 1

Reviewer 1 Report

Comments and Suggestions for Authors

This review aims to analyze the current biochemical, morphological, and functional evidence showing the neuroprotective/neurorestorative effects of Progesterone (P4) against global cerebral ischemia (GCI).

This review is interesting, and it could be useful to the readers.

I suggest modifying the introduction section, presenting the background and the review aims clearly at the end of the paragraph.

Finally, in the conclusion section, I suggest avoiding inserting a picture: usually, this section is important in order to remark on the principal findings of the review suggesting future research lines. I suggest moving the picture to another paragraph; alternatively, I suggest titling the last section "Discussion and Conclusion": in this way, you should discuss the main findings, highlighting the take-home messages and future research lines. Feel free to choose the best way to improve your paper.

Comments on the Quality of English Language

The English is acceptable.

Author Response

Thank you very much for your valuable time to review this work. We sincerely appreciate your constructive comments, as they have contributed to improving the quality of this article.

The detailed response to your comments is presented below, and the corresponding corrections are highlighted in the re-submitted file.           

Comments 1:

This review aims to analyze the current biochemical, morphological, and functional evidence showing the neuroprotective/neurorestorative effects of Progesterone (P4) against global cerebral ischemia (GCI).

This review is interesting, and it could be useful to the readers.

I suggest modifying the introduction section, presenting the background and the review aims clearly at the end of the paragraph.

Response 1:

We agree with this comment.

Following your suggestion, the order of the introduction has been modified. The most pertinent background information now appears at the end of this section, preceding the review's objective.

Comments 2:

Finally, in the conclusion section, I suggest avoiding inserting a picture: usually, this section is important in order to remark on the principal findings of the review suggesting future research lines. I suggest moving the picture to another paragraph; alternatively, I suggest titling the last section "Discussion and Conclusion": in this way, you should discuss the main findings, highlighting the take-home messages and future research lines. Feel free to choose the best way to improve your paper.

Response 2:

Thank you very much, we agree with this recommendation.

We considered the alternative suggestion; therefore, the last section is titled "Discussion and conclusions". In this sense, the main findings were discussed and possible future lines of research were described, as illustrated in Figure 4.

Additional clarifications:

In response to suggestions from other reviewers, section 2, titled “Pathophysiology of global cerebral ischemia and morpho-functional changes” was significantly shortened. In this manner, the length of this section is now concise to avoid misrepresenting the focus of the review.

Reviewer 2 Report

Comments and Suggestions for Authors

The review needs to be completely restructured and shortened. The first part of the manuscript looks like a textbook that has nothing to do with the topic of the review. So, I suggest starting the review with section 2 (guideline recommendations) or even 3 (P4 as a neuroprotective steroid). In the subsequent description of the biochemical and cellular effects of P4, the authors, of course, can provide more information about the biochemical processes in cerebral ischemia, but only if it is necessary to explain the therapeutic effects of P4. In addition, the numbers should explain the mechanisms of action of P4. Therefore, all figures must include P4.

Author Response

Thank you very much for your time in reviewing this manuscript. We greatly appreciate your valuable contributions, which have allowed us to better focus on the objective of our work.

The detailed response to your comments is presented below, and the corresponding corrections are highlighted in the re-submitted file.

Comments:

The review needs to be completely restructured and shortened. The first part of the manuscript looks like a textbook that has nothing to do with the topic of the review. So, I suggest starting the review with section 2 (guideline recommendations) or even 3 (P4 as a neuroprotective steroid). In the subsequent description of the biochemical and cellular effects of P4, the authors, of course, can provide more information about the biochemical processes in cerebral ischemia, but only if it is necessary to explain the therapeutic effects of P4. In addition, the numbers should explain the mechanisms of action of P4. Therefore, all figures must include P4.

Response 1:

Thank you for pointing this out.

In accordance with your suggestion, the manuscript was considerably shortened.

Specifically, section 2, titled “Pathophysiology of global cerebral ischemia and morpho-functional changes” was reduced from 5 and a half pages to just one and a half pages, including only Figure 2, which provides a general overview of the pathophysiology. The other three figures that specifically illustrated distinct pathophysiological mechanisms of damage have been removed. In this manner, the length of this section is intentionally concise to avoid distorting the focus of the review. In this same sense, the final part of the Introduction was modified, and a large portion of the last section “Discussion and conclusions”.

Concerning the illustrations, in Figure 2 referring to the pathophysiology of ischemia, the effect of progesterone on these damage mechanisms was included. In addition, a new figure (Figure 3) was created explaining the effects of progesterone on post-ischemia brain plasticity. Therefore, all the figures in this review now incorporate the role of progesterone, as you rightly indicated.

Reviewer 3 Report

Comments and Suggestions for Authors

1.This research focused on Neuroprotective-neurorestorative effects induced by progesterone on global cerebral ischemia: A narrative review, after check the pubmed, there were some reviews about this topic such as PMID23631651 , 25196185, 36436500,but this manuscript from other perspectives and the content is complete and rich,  so this manuscript was very prospective and significant.

2. Too much contents introduce the GCI, this knowledge are mainly consisted with IRI or apoptosis, neally about 50% of whole manuscript, I adviced to be more concise.

3.Contents 3,4,5 I think were the core content of the entire article, but no tables only 1 Figure, was very inappropriate.

4. Figure 2 about the pathophysiology of GCI, I think following the development of diseases follows a vicious cycle, the cycle maybe two-way, such as mitochondrial dysfunction →excitotoxicity also excitotoxicity→mitochondrial dysfunction.

5.English should be more polish.

Comments on the Quality of English Language

Nearly fine.

Author Response

Thank you sincerely for dedicating time to review this manuscript. Your invaluable contributions have significantly enriched our work.

The detailed response to your comments is presented below, and the corresponding corrections are highlighted in the re-submitted file.           

Comments 1:

This research focused on Neuroprotective-neurorestorative effects induced by progesterone on global cerebral ischemia: A narrative review, after check the pubmed, there were some reviews about this topic such as PMID23631651, 25196185, 36436500,but this manuscript from other perspectives and the content is complete and rich, so this manuscript was very prospective and significant.

Response 1:

We very much appreciate this comment.

Comments 2:

Too much contents introduce the GCI, this knowledge are mainly consisted with IRI or apoptosis, neally about 50% of whole manuscript, I adviced to be more concise.

Response 2:

We agree. Therefore, section 2, titled “Pathophysiology of global cerebral ischemia and morpho-functional changes” was reduced from 5 and a half pages to just one and a half pages, including only Figure 2, which provides a general overview of the pathophysiology. The other three figures that specifically illustrated distinct pathophysiological mechanisms of damage have been removed. In this manner, the length of this section is concise to avoid misrepresenting the focus of the review.

Comments 3:

Contents 3,4,5 I think were the core content of the entire article, but no tables only 1 Figure, was very inappropriate.

Response 3:

We agree with this comment.

To correct this, in Figure 2 referring to the pathophysiology of ischemia, the effect of progesterone on these damage mechanisms was included. In addition, a new figure (Figure 3) was created explaining the effects of progesterone on post-ischemia brain plasticity. Consequently, there are now three figures that are related to the core content of this work.

Comments 4:

Figure 2 about the pathophysiology of GCI, I think following the development of diseases follows a vicious cycle, the cycle maybe two-way, such as mitochondrial dysfunction →excitotoxicity also excitotoxicity→mitochondrial dysfunction.

Response 4:

Thank you so much for the correction. To represent this vicious circle, and avoid confusion, the arrowheads were removed.

Comments 5:

English should be more polish.

Response 5:

We appreciate your observation. The manuscript has been revised and the English writing has been improved.

Round 2

Reviewer 2 Report

Comments and Suggestions for Authors

The authors did a good revision. I have no additional comments.

Reviewer 3 Report

Comments and Suggestions for Authors

As the autors have revised all my concerns, I thinks now is suitable to be accepted and be publicated in this famous journal, but only the editor in chief can make the final decision.